# SARS-CoV-2 Spike and Nucleocapsid Antibody Response in Vaccinated Croatian Healthcare Workers and Infected Hospitalized Patients: A Single Center Cohort Study

**DOI:** 10.3390/v14091966

**Published:** 2022-09-04

**Authors:** Paola Kučan Brlić, Martina Pavletić, Mate Lerga, Fran Krstanović, Marina Pribanić Matešić, Karmela Miklić, Suzana Malić, Leonarda Mikša, Maja Pajcur, Dolores Peruč, Maren Schubert, Federico Bertoglio, Jurica Arapović, Alen Protić, Alan Šustić, Marko Milošević, Luka Čičin Šain, Stipan Jonjić, Vanda Juranić Lisnić, Ilija Brizić

**Affiliations:** 1Center for Proteomics, Faculty of Medicine, University of Rijeka, Braće Branchetta 20, 51000 Rijeka, Croatia; 2Emergency Department, Clinical Hospital Center Rijeka, 51000 Rijeka, Croatia; 3Department of Microbiology and Parasitology, Faculty of Medicine, University of Rijeka, Braće Branchetta 20, 51000 Rijeka, Croatia; 4Institut für Biochemie, Biotechnologie und Bioinformatik, Abteilung Biotechnologie, Technische Universität Braunschweig, Spielmannstr. 7, 38106 Braunschweig, Germany; 5Faculty of Medicine, University of Mostar, Bijeli Brijeg b.b., 88000 Mostar, Bosnia and Herzegovina; 6Department of Anesthesiology, Reanimation, Intensive Care and Emergency Medicine, Faculty of Medicine, University of Rijeka, 51000 Rijeka, Croatia; 7Department of Clinical Medical Science II, Faculty of Health Studies, University of Rijeka, 51000 Rijeka, Croatia; 8Helmholtz Center for Infection Research, Department of Viral Immunology, 38124 Braunschweig, Germany; 9German Centre for Infection Research (DZIF), Partner Site Hannover/Braunschweig, 38124 Braunschweig, Germany; 10Centre for Individualised Infection Medicine (CiiM), Joint Venture of Helmholtz Centre for Infection Research and Hannover Medical School, 30625 Hannover, Germany

**Keywords:** SARS-CoV-2, COVID-19, healthcare workers, COVID-19 vaccine, BNT162 vaccine, hospitalized patients, antibody detection, Spike, Nucleocapsid, Omicron

## Abstract

Studies assessing the dynamics and duration of antibody responses following SARS-CoV-2 infection or vaccination are an invaluable tool for vaccination schedule planning, assessment of risk groups and management of pandemics. In this study, we developed and employed ELISA assays to analyze the humoral responses to Nucleocapsid and Spike proteins in vaccinated health-care workers (HCW) and critically ill COVID-19 patients. Sera of more than 1000 HCWs and critically ill patients from the Clinical Hospital Center Rijeka were tested across a one-year period, encompassing the spread of major SARS-CoV-2 variants of concern (VOCs). We observed 97% of seroconversion in HCW cohort as well as sustained anti-Spike antibody response in vaccinees for more than 6 months. In contrast, the infection-induced anti-Nucleocapsid response was waning significantly in a six-month period. Furthermore, a substantial decrease in vaccinees’ anti-Spike antibodies binding to Spike protein of Omicron VOC was also observed. Critically ill COVID-19 patients had higher levels of anti-Spike and anti-Nucleocapsid antibodies compared to HCWs. No significant differences in anti-Spike and anti-Nucleocapsid antibody levels between the critically ill COVID-19 patients that were on non-invasive oxygen supplementation and those on invasive ventilation support were observed. However, stronger anti-Spike, but not anti-Nucleocapsid, antibody response correlated with a better disease outcome in the cohort of patients on invasive ventilation support. Altogether, our results contribute to the growing pool of data on humoral responses to SARS-CoV-2 infection and vaccination.

## 1. Introduction

The COVID-19 pandemic, caused by the SARS-CoV-2 virus, has been ongoing for more than two years and has resulted in more than 570 million reported cases and over 6.3 million deaths worldwide as of 2 August 2022 (WHO COVID-19 Dashboard). Following the first clinical studies of vaccine efficacy, it was hoped that the administration of vaccines would restrict virus spread [1,2]. Certainly, a temporary plateau of new COVID-19 cases was observed in many countries and the vaccines proved to have a high efficiency in preventing serious COVID-19 illnesses [3]. However, with the emergence of new variants, it became apparent that novel SARS-CoV-2 variants evade neutralizing antibodies generated by infection and vaccination [4,5]. Throughout the entire pandemic, numerous serological studies have assessed the dynamics and duration of antibody responses following SARS-CoV-2 infection or vaccination, with often conflicting findings [6]. Serological tests constitute an essential element for developing strategies for SARS-CoV-2 infection prevention and control. They can complement the diagnostic approach to suspected or confirmed cases of COVID–19; they allow monitoring of seroconversion in the population providing data on the incidence of infection [7,8,9]; they assess dynamics and longevity of antibody responses following vaccination or infection and they can identify some of the asymptomatic cases [10]. Finally, serological studies have a critical role in screening volunteers for vaccine trials or convalescent plasma donation and, with the looming threat of new pandemic waves, in the identification of high-risk population groups that could benefit from additional preventive strategies [11]. Serological tests can be particularly informative and useful in pandemic management, as shown in one model [12], and this is probably even more relevant when applied to a highly exposed cohort of healthcare workers (HCWs). For critically ill COVID-19 patients serology tests may be helpful for clinical management and can contribute to the identification of risk factors associated with worse outcomes [13]. Despite numerous serological studies performed on these cohorts, data on the stability and efficiency of humoral response after vaccination are still scarce, particularly regarding the new Omicron (B.1.1.529) variants. Finally, all of these are affected by multiple factors including age, genetics, lifestyle, and co-morbidities, which is why multiple serological studies are needed.

The aim of this research was to determine the humoral responses in vaccinated HCWs and critically ill COVID-19 patients. To that aim, we have developed ELISA assays based on the Nucleocapsid and Spike proteins, the two most commonly used targets in SARS-CoV-2 serological assays. Spike protein is a structural protein playing a crucial role in the infection of host cells, as its receptor-binding domain (RBD) binds to the angiotensin-converting enzyme 2 (ACE2) receptor, thus initiating viral entry into host cell [14]. Therefore, targeting Spike protein represents the most utilized strategy for vaccine development, and analysis of anti-Spike antibodies allows for monitoring of a vaccination-induced humoral response [2]. Nucleocapsid protein is another major structural protein of the virus that is involved in the transcription and replication of viral RNA and the packaging of the genome into virions and interference with the host’s cell cycle processes [15,16]. Nucleocapsid protein was reported to be highly immunogenic and abundant [17,18] with anti-Nucleocapsid antibodies being more sensitive than the Spike antibodies for early infection detection [19]. In addition, marked humoral immunity to the Nucleocapsid protein reported in COVID-19 patients together with a more conserved aminoacid sequence led to the suggestion that the Nucleocapsid protein is a potential target to be incorporated in future COVID-19 vaccines [20,21,22]. Furthermore, in contrast to antibodies specific for Spike protein, detection of anti-Nucleocapsid antibodies allows for differentiating between vaccinated individuals with or without a history of infection [23,24].

In conclusion, we report development of ELISA assays and their application to early and longitudinal antibody response assessment in a cohort of HCWs following vaccination with BNT162b2mRNA vaccine (Comirnaty; BioNTech/Pfizer, NY, USA), including the period 14–16 months after vaccination. In addition, serological profiling of critically ill, intensive care unit hospitalized (ICU) COVID-19 patients was performed and a correlation between selected demographic factors, outcome and humoral response in these groups was assessed.

## 2. Materials and Methods

### 2.1. Study Population and Serum Processing

#### 2.1.1. Healthcare Workers (HCW)

A total of 1072 HCWs from the Clinical Hospital Center Rijeka were included in the study: 937 were recruited prior to vaccination with the first dose of BNT162b2 (Comirnaty, BioNTech/Pfizer, NY, USA) vaccine (t0), 601 participated in the follow-up prior to the second dose (t1) with an additional 50 new participants included. Six months post vaccination with two doses (t2), 291 of the original cohort samples were collected, plus an additional 13 from cohort t1 and 76 new participants were included. At 14–16 months after vaccination (t3), 19 samples of the original cohort were collected and 1 additional participant was included. All vaccinations were performed with BNT162b2 vaccine. Blood was collected after obtaining informed consent. All participants were older than 18 years and both genders were included. Participants’ demographic data (age, gender) and information whether they had COVID-19 infection were obtained during the sample collection. Additional information about COVID-19 infection, symptoms and booster vaccination was collected through an online questionnaire at various times during collection. The questionnaire between t0 and t2 consisted of 8 questions related to the participants COVID-19 status (yes/no), COVID-19 symptoms (duration of symptoms, type of symptoms, presence of post-covid symptoms, symptoms present > 1 month from disease, hospitalization) and other information (known non-responsiveness to vaccines, immune deficiency diseases/disorders). In total, 449 HCWs answered the online questionnaire. The questionnaire at t3 consisted of 2 questions: COVID-19 status (yes/no) and receival of 3rd booster dose (yes/no, date). A total of 19 HCWs out of 20 answered the questionnaire.

To obtain serum, 10 mL of venous blood was drawn into a serological tube (Vacuette 454067; Greiner Bio-One GmbH, Kremsmünster, Austria) and stored at +4 °C for a few hours, until further processing. Serum was separated by centrifugation for 10 min at 800 g at room temperature (RT). In a BSL2 cabinet, serum was transferred into microtube containing 10% Triton so that the final concentration of Triton was 1% in order to inactivate enveloped virus [25]. The serum was then mixed and left at room temperature for 1 h, aliquoted, and stored at −20 °C until further analysis.

#### 2.1.2. Hospitalized Patients

The cohort consisted of two groups: critically ill COVID-19 patients that required invasive ventilator support and non-critically ill, oxygen-dependent patients without invasive respiratory support. All patients were initially diagnosed with SARS-CoV-2 infection by RT-qPCR testing from nasopharyngeal swabs. The patients were hospitalized 3–14 days following onset of symptoms and blood was collected within the first 24 to 36 h after intubation at the COVID-19 ICU of the Clinical Hospital Center Rijeka. Samples of seventy-four critically ill COVID-19 patients admitted to the COVID-19 ICU of the Clinical Hospital Center Rijeka with the diagnosis of severe COVID-19 and acute respiratory distress syndrome defined by the Berlin criteria were collected. These patients required invasive ventilation support, and ICU specialists followed standardized therapeutic guidelines. These patients were sampled in three periods: November–December of 2020 (33 patients, 0 were vaccinated), April 2021 (23 patients, 0 were vaccinated), and November–December 2021 (18 patients, 2 were vaccinated). Twenty-eight non-critically ill COVID-19 patients were hospitalized with severe, oxygen-dependent COVID-19 during January of 2021. A total of 3 mL of venous blood was drawn into a serological tube and stored at +4 °C for a few hours, until further processing. Serum was separated by centrifugation for 10 min at 800 g at room temperature and then transferred into microtube and stored at −20 °C until further analysis. Prior to ELISA analysis, serum was thawed at RT and inactivated with a 1% final concentration of Triton-X for 1 h. Serum samples from pre-pandemic controls were processed in the same way.

### 2.2. Development of Spike and Nucleocapsid ELISA

For detection of antibodies against SARS-CoV-2 Spike glycoprotein or Nucleocapsid protein in human sera samples, we established in-house ELISA assays. Full Nucleocapsid protein was designed and expressed as His-tagged protein as described in [26]. Recombinant His-tagged Spike protein corresponding to S1-S2 (aa14–1208, with proline substitutions at position 986 and 987 and “GSAS” substitution at the furin site, residues 682–685) of wild type SARS-CoV-2 Wuhan (WT) or to variants of concern Omicron BA.1 (mutations A67V, Δ69–70 (delta HV), T95I, G142D/Δ143–145 (deltaVYY), Δ211 (delta N)/L212I, ins214EPE, G339D, S371L, S373P, S375F, K417N, N440K, G446S, S477N, T478K, E484A, Q493K, G496S, Q498R, N501Y, Y505H, T547K, D614G, H655Y, N679K, P681H, N764K, D796Y, N856K, Q954H, N969K, L981F) and Omicron BA.2 (mutations A27S, D405N, D614G, D796Y, E484A, G142D, G339D, H655Y, K417N, L24del, N440K, N501Y, N679K, N764K, N969K, P25del, P26del, P681H, Q493R, Q498R, Q954H, R408S, S371F, S373P, S375F, S477N, T19I, T376A, T478K, V213G, Y505H) were produced baculovirus-free insect cell Hive Five expression system as described before [26,27]. For detection of antibodies against SARS-CoV-2 receptor-binding domain (RBD) corresponding to Wuhan-Hu-1 (WT) virus, His-tagged RBD (residues 319 to 541 of Spike protein) was expressed using a pCAGGS expression vector as described in [28]. His-tagged RBD of Omicron BA.1.1.529 (mutations G339D, S371L, S373P, S375F, K417N, N440K, G446S, S477N, T478K, E484A, Q493R, G496S, Q498R, N501Y, Y505H) was produced as described in [27]. High-binding ELISA plates (MICROLON^®^ High Binding, Greiner Bio-One, Kremsmünster, Austria) were coated with indicated proteins at 2 µg/mL concentration in carbonate/bicarbonate coating buffer pH 9.6 by incubation overnight at 2–8 °C. The plates were then washed two times with tap water [29] and saturated with blocking buffer (ChonblockTM, cat.no. 90681, Chondrex, Inc, Woodinville, WA, USA) [30,31] according to manufacturer’s instructions. After 1 h blocking, plates were washed 3 times with washing buffer (PBS with 0.05% Tween 20) and serum samples diluted 1:100 in the ChonblockTM buffer were added and incubated for 2 h at room temperature. Following incubation with serum and subsequent washing, secondary antibody, peroxidase-conjugated anti-human IgG Fc (Sigma, A0170-1ML, Burlington, MA, USA) was added in ChonBlockTM Detection Antibody Dilution Buffer diluted to 1:2000 (Spike ELISA) or 1:4000 (Nucleocapsid ELISA). Following 1 h incubation and washing, a colorimetric reaction was performed by adding TMB solution (Abcam, cat.no. ab171523, Cambridge, UK) for 3–5 min on RT. The reaction was stopped by addition of Stop solution according to manufacturer’s instructions (Abcam, cat.no. ab171529, Cambridge, UK) and the absorbance was measured using a TriStar LB 941 multimode microplate reader with the wavelength set at 450 nm. The absorbance of the blank wells was subtracted from all standard and sample absorbances. ELISAs using RBD target antigens were performed as described [28], except for the last step, where we used TMB solution to perform colorimetric reaction.

Serum samples were tested in monoplicate. The cut-off values for individual antigen were calculated by adding 3 standard deviations to mean ELISA ODs of pre-pandemic sera samples [32,33]. Signal to cut-off ratio (S/Co) was calculated for each sample and obtained S/Co was considered positive if it was >1.1 [34]. ELISAs for cohort of HCW samples collected prior to vaccination (*n* = 937) were performed once after which all samples that tested positive (S/Co > 1.1), borderline samples (S/Co ≥ 0.8–1.1), and several randomly selected negative samples (S/Co < 0.8) were re-tested. All other samples were tested at least two times and the average S/Co was calculated. To assure the validity of the results each assay included 3–5 commercially available, validated SARS-CoV-2 IgG negative sera (product # DSPA 4.1.9.13.1, in.vent Diagnostica GmbH, Hennigsdorf, Germany) and a reference pool of converted serum samples (EURM-017, Joint research center, European commission). The binding capacity of vaccinees’ antibodies to Spike protein of Omicron VOC in relation to WT Spike protein was calculated by following the formula: % of binding capacity to Omicron = (S/Co ratio obtained for Omicron * 100)/S/Co ratio obtained for WT, for each sample.

### 2.3. Statistics

Quantitative variables were expressed as the median and compared using Student’s *t*-test when comparing two groups or analysis of variance (ANOVA) followed by Tukey’s multiple comparison test for analysis of >2 groups. Mixed effect analysis with Holm–Sidak’s multiple comparisons test was used for comparison of >2 groups for paired samples. Statistical analyses were performed using the GraphPad Prism software (version 8.1.0, GraphPad, San Diego, CA, USA). *p* values < 0.05 were considered significant.

## 3. Results

### 3.1. Development and Validation of ELISA Methods for Screening of Anti-Spike and Anti-Nucleocapsid Antibody Response

In this study, we developed indirect ELISA assays to detect SARS-CoV-2 specific IgG antibodies in serum samples, using several different antigens: the full-length Spike of WT SARS-CoV-2 and variants of concern Omicron BA.1 and Omicron BA.2; the RBD of WT and Omicron BA.1.1.529 variant; and the full-length Nucleocapsid protein. We first determined the background by calculating cut-off values for each target antigen using pre-pandemic sera, as described in the methods section. Using this method, we identified only 1 false positive out of 77 tested pre-pandemic samples for reactivity to WT full length Spike, 2 out of 100 were positive on Nucleocapsid protein, 1 out of 44 reactive to Omicron BA.1 and BA.2 Spike and 1 out of 44 false positive for both RBD WT and RBD Omicron BA.1.1.529 ELISAs (Appendix A). Positive control serum tested positive in all ELISAs except the RBD Omicron. Overall, our results indicated a high specificity of the established ELISAs. Values above the cut-off observed for two pre-pandemic samples are probably due to cross-reactive antibodies, perhaps induced by a related coronavirus infection [35].

For WT Spike ELISA we have assessed assay sensitivity. A total of 56 randomly selected pandemic serum samples were tested by the Teaching Institute of Public Health of Primorsko-goranska county, Croatia, using reference assay SARS-CoV-2 IgG II Quant that detects anti-spike RBD Abs (Abbott Alinity, Abbott, Chicago, IL, USA)). The comparison showed a high correlation of the two assays: more than 90% of samples (51/56) received the same classification (S positive or negative) and a strong correlation between ELISA S/Co values used in our method and AU/mL concentrations used in the reference method was observed (Appendix A). Four out of five samples that were classified as negative in our assay had lower concentrations (less than 200 AU/mL in SARS-CoV-2 IgG II Quant assay), suggesting a somewhat lower sensitivity of our WT Spike ELISA (data points shown in dark gray in Appendix A). One out of five samples was classified as positive in our assay but negative in SARS-CoV-2 IgG II Quant assay (data point shown in light gray in Appendix A).

### 3.2. Demographic and Clinical Characteristics of the HCW Cohort

The cohort included HCWs from Clinical Hospital Center Rijeka. In Croatia, HCWs and the elderly were populations that were prioritized for vaccination. Therefore, HCWs were among the first that received the SARS-CoV-2 vaccine. The timeline of serum sample collection in relation to the COVID-19 pandemic waves in Croatia from the first confirmed case (25 February 2020) is shown in Figure 1A.

At the first time point (t0), prior to vaccination with the BNT162b2 vaccine (Comirnaty, BioNTech/Pfizer, NY, USA), we have collected the sera of 937 HCWs. At 3 weeks after the first dose (t1) and prior to the 2nd dose, 651 sera samples were collected. Six months post vaccination with two doses (t2), 380 samples were collected. Finally, 14–16 months after vaccination with 1st dose (t3), 20 samples were collected (Figure 1). Participants’ demographic data (age, gender) and information whether they had COVID-19 were obtained during the sample collection and are shown in Table 1. Additional information about COVID-19 infection, symptoms and booster vaccination was collected through an online questionnaire between t0 and t2 and at t3 (14–16 months after vaccination). For all time points, majority of the participants were women and the age of the participants ranged from 19 to 70, with an average of around 46 years. Out of 160 HCWs that reported COVID-19 prior to, or in the 6 months after vaccination, 82 answered the questionnaire. Amongst them, seven (8.5%) reported no symptoms at all, 18 reported symptoms lasting up to 3 days (24%), 30 reported symptoms up to 7 days (40%), 16 reported symptoms up to 14 days (21.3%), and 11 reported symptoms for more than 14 days (14.7%). The most commonly described symptoms included increased body temperature, tiredness, cough, or muscle pain. None of the HCWs reported hospitalization. At t3, 14–16 months post vaccination, 7/19 (36%) HCWs reported having had COVID-19 in the period from the last sampling and 14/19 HCWs reported having received the 3rd booster dose.

### 3.3. Longitudinal Assessment of Anti-Spike and Anti-Nucleocapsid Antibody Response in HCWs

Anti-Spike and anti-Nucleocapsid antibody response in HCWs was assessed at four time points (Figure 1A and Figure 2A). Anti-Spike antibodies were detected in 9.93% (93/937) of HCWs before vaccination (t0) (Figure 2B). Before administration of the second vaccine dose (t1, 3 weeks after receiving the first dose), seroconversion was detected in more than 85% (566/651) of participants with a further increase in the seroconversion rate to 97% (369/380) 6 months from the 2nd vaccination dose (t2), confirming the benefit of a second vaccine dose. When only seroconverted individuals were compared, sustained anti-Spike antibody S/Co ratio, above the antibody level 3 weeks after the first vaccine dose, was observed in the six-month period, with a further increase 14–16 months after vaccination (t3) (Figure 2C). A similar result was obtained when we compared more than 200 paired samples (Figure 2D). This analysis also included HCWs that self-reported having had COVID-19, which might contribute to increase of the anti-Spike antibody levels. Nevertheless, a similar result was observed when HCWs with a history of COVID-19 infection (identified by anti-Spike or anti-Nucleocapsid seroconversion) were excluded from the paired samples (Figure 2E), suggesting that this sustained response is due to vaccination. Despite the overall higher levels of antibodies at 6 months after the second dose in comparison to 3 weeks post first dose, it should be noted that at 6 months more than 35% of the paired cohort had a reduced anti-Spike S/Co (Figure 2D, red lines), suggesting a stronger decrease of anti-Spike antibodies in a part of the cohort. A similar reduction in antibody titers was observed after the exclusion of individuals with a history of infection (Figure 2E, red lines). At the last time point, 14–16 months after initial sampling, anti-Spike antibodies were detected in all tested individuals (Figure 2B) and levels of antibodies were increased when compared to the 6-months post-vaccination time point (Figure 2C,D). However, at this time point all individuals either reported history of COVID-19 or received a booster dose, or both, explaining increased antibody levels as compared to the 6-month time point (Table 1).

To assess the effect of the history of COVID-19 on anti-Spike antibody levels in vaccinees, we first compared anti-Spike S/Co between all t2 samples based on their COVID-19 self-reporting. Indeed, participants reporting previous COVID-19 infection had higher levels of anti-Spike antibodies (Figure 2F). To further elucidate the effect of SARS-CoV-2 infection on anti-Spike antibody response, we stratified the HCW cohort into 12 groups based on the timing of infection (Figure 2A). For this stratification, we used only samples of HCWs involved in the study from t0. Infection status was determined by participants’ self-report on positive PCR testing and/or by detection of anti-Nucleocapsid antibodies at indicated timepoints (Figure 2A). Sustained anti-Spike antibody S/Co ratio above the levels observed after the first vaccination dose in non-infected individuals was observed in the period of 6 months (Figure 2G). A further increase was observed at 14–16 months after primary vaccination probably due to the booster vaccination that occurred in the indicated period (Figure 2A), supporting vaccination efficacy (groups G1, G3, G6, G10; Figure 2G). Sustained anti-Spike response was also observed in individuals that were both vaccinated and infected (groups G2, G5, G9, G12; Figure 2G). Furthermore, infection preceding or following vaccination increased anti-Spike antibody levels in comparison to the non-infected group in the majority of time points tested (groups G2 vs. G1, G5 vs. G3 or G9 vs. G6; Figure 2G). In the much smaller cohort analyzed at 14–16 months since vaccination, the analysis revealed no difference in anti-Spike antibody response in people that reported having had COVID-19 and received a 3rd (booster) dose in comparison to HCWs that were only vaccinated three times or had contracted SARS-CoV-2 virus and did not receive a booster dose (Appendix A).

Finally, to assess if demographic factors affect antibody response to vaccination and infection, we compared the S/Co of anti-Spike antibodies on the entire cohort at the time point of 6 months (t2), based on the participants age (Figure 2H) and gender (Figure 2I). Similar anti-Spike response was observed between analyzed age subgroups. Similarly, there was no difference in anti-Spike response between men and women, suggesting no influence of gender, although disproportionate numbers of male and female participants in our cohorts have to be taken into account. The same findings were observed on the smaller cohort analyzed at 14–16 months since vaccination (Appendix A). Given the continuous emergence of new SARS-CoV-2 variants escaping humoral immunity, sera collected at 6 months post vaccination (t2) were also tested for their recognition of Spike protein encoded by the Omicron subvariants BA.1 and BA.2 (Figure 3). In contrast to WT Spike protein (Figure 2B) for which 97.11% of sera samples of vaccinated HCWs were positive, 75.26% and 80.79% sera samples were positive for BA.1 and BA.2 Omicron subvariants, respectively (Figure 3A). Furthermore, the binding capacity of anti-Spike antibodies was significantly reduced as compared to the binding to WT protein for most of the analyzed sera (Figure 3B,C), suggesting impaired sensitivity of the Omicron variants to antibodies. This was even more pronounced for Omicron RBD that escaped antibody recognition in more than 50% of analyzed participants (Figure 3D) and that showed a substantial decrease in the binding capacity of anti-Spike antibodies (Figure 3E). When stratifying participants in t2 based on the presence and timing of previous COVID-19 infection (Figure 2A), the capacity of anti-Spike antibodies to bind Omicron BA.1 and BA.2 Spike proteins and RBD protein was significantly reduced in most of the groups when compared to WT Spike and RBD protein, respectively, except for the group with the most recent COVID-19 infection (group G7, Figure 3F). Additionally, the reduction of binding to Spike protein encoded by the Omicron subvariants BA.1 and BA.2 was most pronounced for HCWs that were not infected (group G6, Figure 3F). In line with that, correlation analysis of anti-WT Spike S/Co vs. anti-Omicron S/Co suggested the strongest correlation between groups that had more recent infection, while the lowest correlation was observed, for all variants tested, in the group that had no evidence of prior infection at t2 (Appendix A). These data suggest that the combination of both previous infection and vaccination could provide better protection against Omicron and that if the infection was more recent protection should be better.

The analysis of the binding capacity was also performed for samples collected 14–16 months after vaccination (Appendix A). Samples collected on this time point had also overall reduced binding capacity to Omicron as compared to either WT full length Spike protein (Appendix A) or WT RBD (Appendix A).

In addition to testing the anti-Spike response to vaccination, we also tested the antibody response to highly immunogenic Nucleocapsid protein to assess the incidence of infected HCWs prior to vaccination and the kinetics of infection induced anti-Nucleocapsid antibody response. A slightly higher frequency of seroconverted HCWs (12.06%) was observed using Nucleocapsid ELISA as compared to anti-Spike ELISA (9.93%) (Figure 4A). This could be in part due to higher cross-reactivity of the anti-Nucleocapsid ELISA (Appendix A), higher affinity of anti-Nucleocapsid antibodies rendering this assay more sensitive, or higher levels of anti-Nucleocapsid antibodies. In the samples collected at t2, excluding participants who had anti-Spike antibodies at t0 or anti-Nucleocapsid antibodies at t0 and t1, we detected 4% of anti-Nucleocapsid positive, vaccinated HCWs, probably representing breakthrough infections (Figure 4B). Finally, when analyzing the kinetics of the anti-Nucleocapsid antibodies for paired samples collected in at least two time points, we observed a significant decline in the anti-Nucleocapsid antibody levels in a period of 6 months of sampling, which is in contrast to anti-Spike antibody response following vaccination (Figure 4C).

Furthermore, to assess if there is a difference in anti-Nucleocapsid antibody response in relation to demographic parameters, we stratified seroconverted individuals identified prior to vaccination and compared anti-N response between different age groups and genders (Figure 4D,E). Here, similar to what was observed for anti-Spike response in vaccinated individuals, we did not observe any differences in relation to age or gender of anti-Nucleocapsid response in t0 seroconverted individuals.

Overall, our data on HCWs suggested an increase of the anti-Spike antibody levels in the majority of the tested individuals 6 months after vaccination, while the anti-Nucleocapsid antibody levels waned in SARS-CoV-2 infected individuals.

### 3.4. Demographic and Clinical Characteristics of the Hospitalized Patients’ Population

Since the severity of the disease has been identified as one of the most critical factors in the humoral response to SARS-CoV-2, we collected serum samples from hospitalized patients with severe symptoms that were placed either on oxygen supplementation or invasive ventilation support (IVS) (Table 2) and analyzed their anti-Spike or anti-Nucleocapsid antibody response. The study population included 102 patients admitted to the COVID-19 ICU of the Clinical Hospital Center Rijeka as described in the Materials and Methods and partially in [36]. The cohort included 28 non-critically ill COVID-19 patients that were hospitalized and oxygen-dependent and 74 critically ill COVID-19 patients that required IVS (Table 2). Both groups consisted mostly of male patients (72%), with ages ranging from 40 to 89 years.

The critically ill COVID-19 patient cohort undergoing IVS had a high mortality rate with no substantial differences in duration of the IVS between survivors and non-survivors but a higher sequential organ failure assessment (SOFA) severity score at admission in the non-survivor group (Table 3, Figure 5A), in accordance with a previous report [36]. Of note, only two patients out of 18 that were sampled in November/December 2021 that were on IVS have been vaccinated.

### 3.5. Anti-Spike and Anti-Nucleocapsid Antibody Response in Hospitalized Patients

First, we analyzed the frequencies of patients with anti-Spike (Figure 5B) or anti-Nucleocapsid (Figure 5C) antibodies above the cut-off value. The majority of the hospitalized patients had detectable antibodies to Spike or Nucleocapsid. When stratifying patients based on treatment, we observed similar seroconversion rates between patients that required supplementary oxygen but without the need for IVS and those on IVS (68% vs. 60% for Spike and 75% vs. 72% for Nucleocapsid, Figure 5B,C). Similar to what was observed for the HCW cohort, higher seroconversion rates were observed for the Nucleocapsid than for Spike antibodies (73% vs. 62%).

Next, we compared the levels of anti-Spike antibodies in seroconverted hospitalized patients to the cohort of HCW identified as anti-Spike seroconverted prior to vaccination (Figure 6A). We observed statistically higher levels of anti-Spike antibodies in hospitalized patients (Figure 6A) and no statistical differences between the two groups of hospitalized patients (Figure 6B). As in the vaccinated HCW cohort, levels of the anti-Spike antibodies in hospitalized seroconverted patients were not affected by age or gender (Figure 6C,D). When stratifying patients that were on IVS (Table 3) based on the outcome, we observed a trend suggesting fewer anti-Spike antibodies in the non-survivor group (*p* = 0.06, Figure 6E). Furthermore, we observed higher survival in the anti-Spike seroconverted group than in the patients with no anti-Spike antibodies detected (41% vs. 13%; Figure 6F). In line with that, we also observed a weak but statistically significant negative correlation when comparing Spike antibody S/Co to patients SOFA score (Pearson r = −0.4; Appendix A) or Acute Physiology and Chronic Health Evaluation (Apache II) score (Pearson r = −0.3; not shown), both indicating disease severity and predictive of the mortality.

The same analyses were performed for anti-Nucleocapsid seroconverted patients (Figure 7). First, we compared their anti-Nucleocapsid antibody response to the HCW cohort (Figure 7A) that we identified to be anti-Nucleocapsid seroconverted prior to vaccination (Figure 4A). As for anti-Spike antibodies, higher levels of the anti-Nucleocapsid antibodies were observed in the hospitalized cohort compared to HCWs with a history of mild COVID-19. Furthermore, stratification of patients based on the severity of disease revealed no differences in the anti-Nucleocapsid S/Co between the two groups (Figure 7B), nor were there any statistical differences related to age or gender (Figure 7C,D). Finally, anti-Nucleocapsid antibody scoring did not reveal any differences in relation to the survival of the patients that were on IVS and showed a high mortality rate (*p* = 0.27, Figure 7E). In line with that, differences in frequencies of survivors in relation to anti-Nucleocapsid seroconversion were less pronounced (32% in seroconverted vs. 24% in no Nucleocapsid antibodies group) than those observed in relation to anti-Spike seroconversion (Figure 7F), and there was no statistical correlation between Nucleocapsid antibody S/Co and patients SOFA score (Appendix A).

Overall, our data points to stronger antibody response to major structural proteins in hospitalized COVID-19 patients as compared to HCWs with prior infection and supports the association of weaker humoral response with the severity of the disease and the outcome in the critically ill.

## 4. Discussion

Since the beginning of the pandemic and especially since the availability of the first anti-SARS-CoV-2 vaccine, the main question was the durability, robustness, and efficacy of the elicited immune responses against the parental strain. The answer to this question became even more important as new variants of concern, many bearing mutations in the main target of most vaccines—protein Spike gene region—evolved. Immune responses to infection and/or vaccines are influenced by a multitude of factors—from genetics to age, gender, environment, and other factors, which is why investigations of immune responses to the novel virus remain and will continue to be of major importance for public health and vaccine policy planning for years to come.

Therefore, to answer this need, the objective of our research was to use Spike and Nucleocapsid ELISA to study the humoral responses to vaccination or natural SARS-CoV-2 infection in highly exposed and highly affected cohorts of HCWs and hospitalized patients, respectively. Analysis of the anti-Spike and anti-Nucleocapsid antibody response in HCWs prior to vaccination identified approximately 10% and 12% seroconverted HCWs, respectively. A slightly higher frequency of anti-Nucleocapsid positive individuals could be partially explained by lower specificity of Nucleocapsid ELISA but could also be due to a reported higher sensitivity of anti-Nucleocapsid antibodies in the detection of recent infection [19,37]. Nevertheless, observed frequencies are similar to those reported for the same period of sampling (after the 2nd wave) for HCWs in other European countries, occasionally reaching up to 30% [38,39,40]. In comparison to what has been observed for the general Croatian population, frequencies observed here for HCWs were lower, which is probably attributable to enhanced preventive measures used by HCWs [41]. Following vaccination, already after the first dose and more pronounced following the second, we observed an increase in the anti-Spike antibody levels, confirming the efficacy of vaccination, as also reported by others [42]. VOCs Delta and especially Omicron are characterized by changes in Spike structure, which reduce efficiency of antibodies generated by vaccination. We have observed approximately 4% of new N-seroconverted vaccinated individuals in a period of 6 months following vaccination, coinciding with the appearance of the Delta variant. At the last time point, 14–16 months post vaccination, 7/19 (36%) HCWs reported to have had COVID-19 between November 2021 and March 2022, coinciding with the period after the Omicron variant became dominant. When sera collected at 6 months post-vaccination were tested for their recognition of recombinant Spike or RBD protein corresponding to variants BA.1 or BA.2 Omicron, a reduction in their binding capacity, as compared to the binding to WT protein, was observed, again in accordance with other published results [43,44]. In particular, when sera was retested using RBD-Omicron, more than 50% of the analyzed cohort lost positivity and showed a substantial decrease in anti-Spike antibodies, as previously observed [45,46,47]. The same findings were observed for samples collected 14–16 months from initial vaccination, a cohort that reported having COVID-19 or that received a booster dose and suggesting that new vaccine formulations should consider new variants of immune escape mutations [48]. However, this cohort is quite small and we lack information about the type of VOC infecting a portion of our cohort. As most of the participants from this cohort were infected at the beginning of the Omicron wave, it is possible they contracted an earlier VOC.

The stability of anti-Spike antibody response upon vaccination or infection has been reported as somewhat controversial [49,50], but the majority of studies have shown an overall decrease in antibody levels over time [51,52,53,54]. Some studies suggest a significant and rapid decrease in the humoral response to the BNT162b2 vaccine 6 months post-vaccination and yet a high efficacy of persisting antibodies in SARS-CoV-2 neutralization [42,55,56]. Others showed a sustained response over 6 months in immunocompromised cancer patients [57] or the healthy cohort for up to 9 months [58]. Finally, there also exists reports showing an increase in the titers of IgG antibodies against Spike protein two months since first vaccination [59]. Our results at 6 months post-vaccination show that an anti-Spike antibody response in approximately 70% of participants was sustained or was slightly higher than the level of antibodies at 3 weeks after the first vaccination dose. The drawback of our study is that we did not assess the level of antibodies in the first several weeks after vaccination with the second dose, i.e., the timepoint when the levels of vaccine-induced antibodies is the highest [55,60]. If this analysis was included, we expect that waning of antibody response would be pronounced in our case as well. Most of the studies showing rapid antibody waning used RBD as ELISA target protein [42,55], whereas we used full Spike (S1-S2) protein to assess longitudinal stability of antibody response. Accordingly, previous studies suggested that anti-RBD antibody titers seem to decline faster than anti-Spike antibody titers [61]. Increased antibody levels in samples at 14–16 months following vaccination are attributable to the fact that these individuals were either vaccinated with a booster dose and/or had COVID-19 in the meantime. Similar results were obtained by stratifying groups based on the timing of infection. Namely, the combination of infection and vaccination in a majority of cases led to increased levels of antibodies as compared to vaccination only. We observed some exceptions (Figure 2G), however, this is probably due to the small number of participants in certain groups. On the other hand, follow-up of the anti-Nucleocapsid response for paired samples collected in at least two time points revealed a significant decline in the anti-N antibodies in a period of 6 months. This is in line with numerous publications suggesting that anti-N antibodies are not detectable in the majority of patients who recovered from COVID-19 6 months after the onset of symptoms [62,63,64,65]. These findings imply that Nucleocapsid protein may not be a useful target in long-term serological population studies.

In accordance with the recommendations regarding age and gender data when assessing vaccination responses [66,67], we performed age and gender stratification of our data. We did not observe any age-related differences in Spike antibodies at 6-months’ time point of the vaccinated cohort or in anti-Nucleocapsid antibodies in seroconverted individuals identified prior to vaccination. However, it should be noted that our cohort of vaccinated HCWs did not include participants over the age of 80, reported to have attenuated vaccination- or natural infection-induced antibody response [68,69,70,71]. Similarly, we analyzed the humoral response in relation to gender. Sexual dimorphism has already been described in the context of innate and adaptive immune responses in general and in the context of susceptibility to COVID-19 infection [72]. Meta-analysis suggested increased efficacy in vaccination-induced protection in men compared to women [73] while several other publications reported higher vaccination-induced antibody levels in women [74,75]. Several publications reported no gender related differences [71,76], following immune responses elicited by virus-infection. Our analysis also did not reveal any statistical gender-related differences in anti-Spike antibody response in the vaccinated cohort at 6-months’ time point or in anti-Nucleocapsid antibodies in seroconverted individuals prior to vaccination. However, our cohort contained more women than men.

Several studies suggested that the additional antigen exposure due to natural infection boosts the quantity of vaccination-induced humoral immune response [70,77,78]. In line with that, when stratifying the data of 6-months’ vaccinees based on previous COVID-19 infection, we observed higher anti-Spike levels in HCWs that have self-reported to have had COVID-19 prior to sample collection, indicating that an increased number of exposures to SARS-CoV-2 antigen(s) enhances the antibody responses, as reported by other [79]. Previous infection was also beneficial in combination with vaccination against VOCs, as it provided better antibody binding of Omicron BA.1 and BA.2 Spike and RBD proteins. This is especially evident if the infection was more recent, altogether arguing that the combination of vaccination and infection provides enhances protection against SARS-CoV-2, as also suggested by others [80,81].

The severity of the COVID-19 disease has been identified as one of the most critical factors in the humoral response to SARS-CoV-2 [82,83,84,85]. In general, severe infections induce higher levels of antibodies, probably due to higher viral loads at the onset of symptoms [86]. Accordingly, the majority of the hospitalized patients we analyzed have seroconverted, as observed by others [87,88]. Furthermore, when we compared the scoring of antibodies in seroconverted patients to the cohort of seropositive HCWs identified prior to vaccination, we observed higher levels of both anti-Spike and anti-Nucleocapsid antibodies in hospitalized patients. Similar findings were observed when comparing hospitalized COVID-19 patients and healthcare workers four months post-infection [89,90] or by comparison of antibody responses in different severity categories of patients being treated for COVID-19 [37,91,92], supporting the association between severity of the disease and humoral response. However, since both cohorts were not sampled in the same period following infection, we cannot exclude that the observed differences would be less pronounced if these data were available. On the other hand, in favor of our results, another study identified that patients admitted to the intensive care unit had significantly higher antibody levels at all intervals after 5 days following symptom onset as compared to mild cases [93]. When analyzing antibody response for patients that were on IVS based on the outcome of hospitalization, we observed a trend toward higher anti-Spike levels in the survivors, reinforcing the findings of others [94,95]. This was not the case for anti-Nucleocapsid antibody levels that were not different between survivors and non-survivors, probably due to non-neutralizing capacity of these antibodies.

This study has several limitations, besides the general drawbacks of serological testing [96]. First, we did not have the possibility to test the neutralization capacity of antibodies. Although it has been suggested that levels of Spike antibodies correlate with neutralization capacity [83], this finding was corroborated by anti-RBD antibody testing while our study utilized full length Spike protein. Furthermore, our Nucleocapsid ELISA showed a small degree (2%) of cross-reactivity in pre-pandemic samples, suggesting some of the results might have been misinterpreted as false positives, although it has been suggested that coronavirus cross-reactivity plays an important part in broadening immune responses toward COVID-19 [97,98]. Finally, despite the significant number of participants enrolled, as a single-center study, our findings may not be generalizable to other geographic locations and patient populations.

Despite these limitations, this research contributes to the understanding of antibody response following vaccination or infection, in particular in the context of emerging variants of concerns and important cohorts of highly exposed HCWs or critically ill, hospitalized patients. In addition, it speaks in favor of vaccination as the main preventive measure against this virus and highlights the importance of continuous serological testing as new VOCs and possibly new vaccine variants arrive.

## Figures and Tables

**Figure 1 viruses-14-01966-f001:**
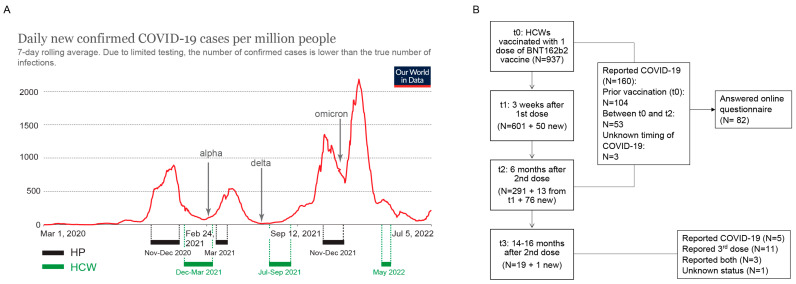
Timeline of serum samples collection and flowchart of HCW cohort enrolment. (**A**) Intervals of sera collection for hospitalized patients (HP) and healthcare workers (HCW) in relation to COVID-19 pandemic waves in Croatia. (**B**) Flowchart of HCWs enrolment, follow-up and COVID-19 status. Image in (**A**) adjusted from Our World in Data.

**Figure 2 viruses-14-01966-f002:**
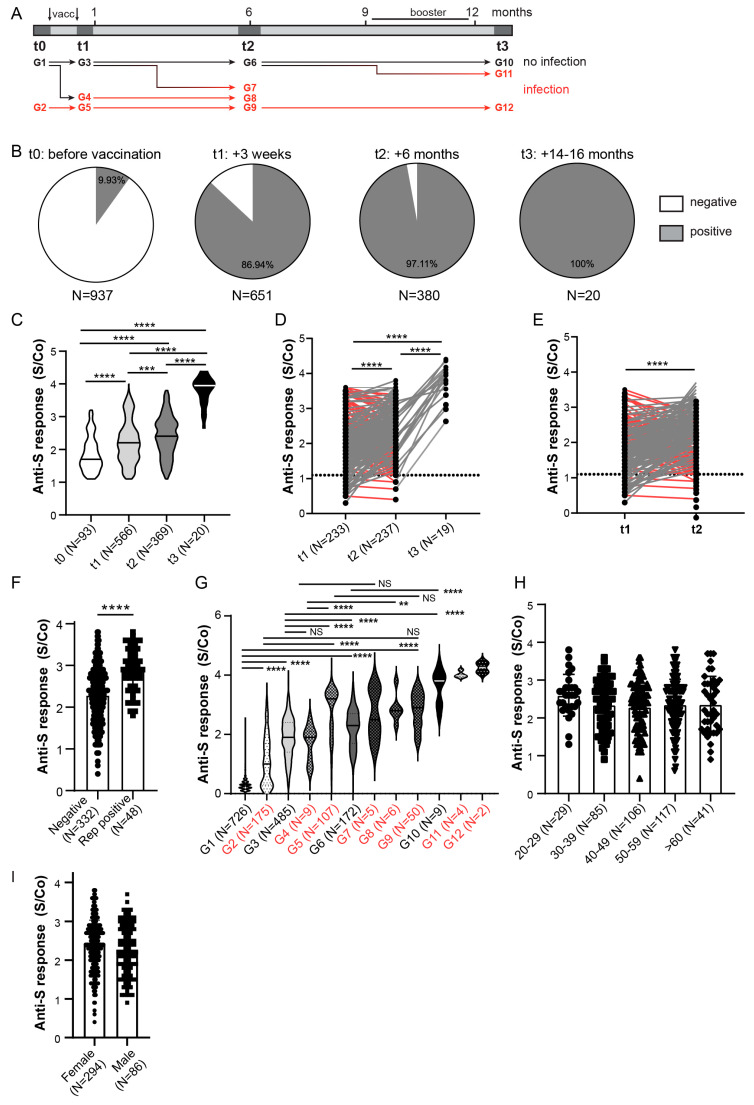
Anti-Spike antibody response in HCW cohort. (**A**) Schematic of serum collection and groups analyzed in (**G**) in relation to timing of SARS-CoV-2 infection. (**B**) Anti-Spike (anti-S) seroconversion rates (above cut-off value) in HCWs at different sera collection time points. (**C**) Comparison of the anti-Spike antibody response between entire cohort of seroconverted individuals (S/Co > 1.1). (**D**) Comparison of the anti-Spike antibody response between all paired samples or (**E**) paired samples that were not Spike seroconverted prior to vaccination or Nucleocapsid protein seroconverted at any time point. Red lines samples displaying reduced anti-S response between t1 and t2 analyses. (**F**) Comparison of anti-Spike antibody S/Co 6-months post vaccination (t2) stratified to participants’ self-reported PCR confirmed COVID-19 infection. (**G**) Comparison of the anti-Spike antibody response in relation to timing of infection assessed by Nucleocapisd seroconversion and self-reported, PCR confirmed COVID-19 infection. (**H**) Comparison of anti-Spike antibody S/Co 6 months post vaccination (t2) stratified to participants’ age or (**I**) gender. S/Co, Signal to cut-off ratio; dashed line indicates cut-off value (S/Co = 1.1); red lines in D and E show samples with reduction of anti-Spike S/Co; red lines and groups in A and G indicate samples with COVID-19 infection. Statistical tests: ordinary one-way ANOVA followed by Tukey’s multiple comparison test (**C**,**G**), mixed effect analysis with Holm-Sidak’s multiple comparisons test (**D**), paired two-tailed *t*-test (**E**,**H**), *p* value **** *p* < 0.0001, *p* value *** and ** *p* < 0.05, NS-not significant.

**Figure 3 viruses-14-01966-f003:**
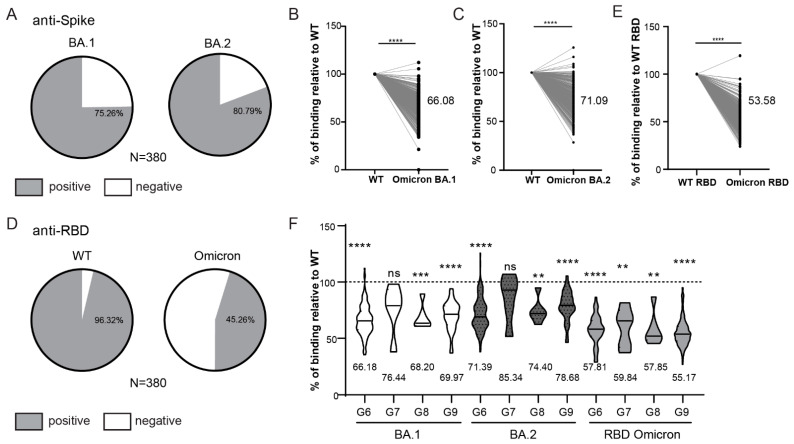
Reactivity of vaccinated HCW’s sera to Omicron Spike variants. (**A**) Rates of 6-month vaccinees (t2) samples reactive to Omicron BA.1 and BA.2 Spike protein. (**B**) Comparison of the intensity of t2 serum binding to Omicron BA.1 and (**C**) BA.2 relative to binding to WT Spike protein. (**D**) Rates of t2 samples reactive to WT RBD and Omicron RBD protein. (**E**) Comparison of the intensity of t2 serum binding to Omicron RBD protein relative to binding to WT RBD protein. (**F**) Comparison of the relative intensity of t2 serum binding to Omicron proteins in relation to the timing of infection (see Figure 2A). S/Co, Signal to cut-off ratio. Statistical tests: paired *t*-test (**B**,**C**,**E**), *p* value **** *p* < 0.0001, *p* value *** and ** *p* < 0.05, ns-not significant.

**Figure 4 viruses-14-01966-f004:**
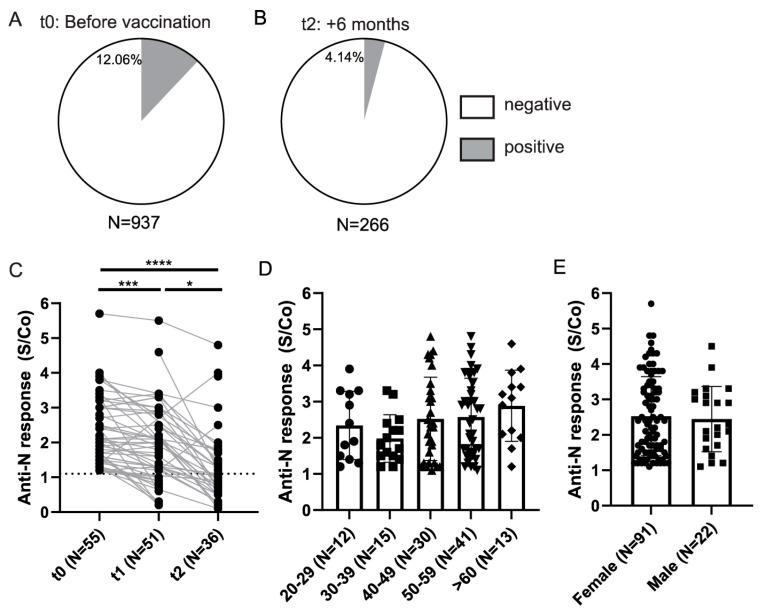
Anti-Nucleocapsid antibody response in HCWs. (**A**) Anti-Nucleocapsid (anti-N) seroconversion rates in HCWs prior to vaccination and (**B**) 6 months after the full vaccination (t2) (S-positive individuals at t0 and N-positive at t0 and t1 were excluded). (**C**) Comparison of the anti-Nucleocapsid antibody S/Co between paired seroconverted samples. (**D**) Levels of anti-Nucleocapsid antibodies in sera of seroconverted individuals at t0 stratified on the basis of age and (**E**) gender. S/Co, Signal to cut-off ratio; dashed line indicates cut-off value (S/Co = 1.1). Statistical test: Mixed-effects analysis with Tukey’s multiple comparisons test (**C**), *p* value **** *p* < 0.0001, *p* value *** and * *p*<0.05.

**Figure 5 viruses-14-01966-f005:**
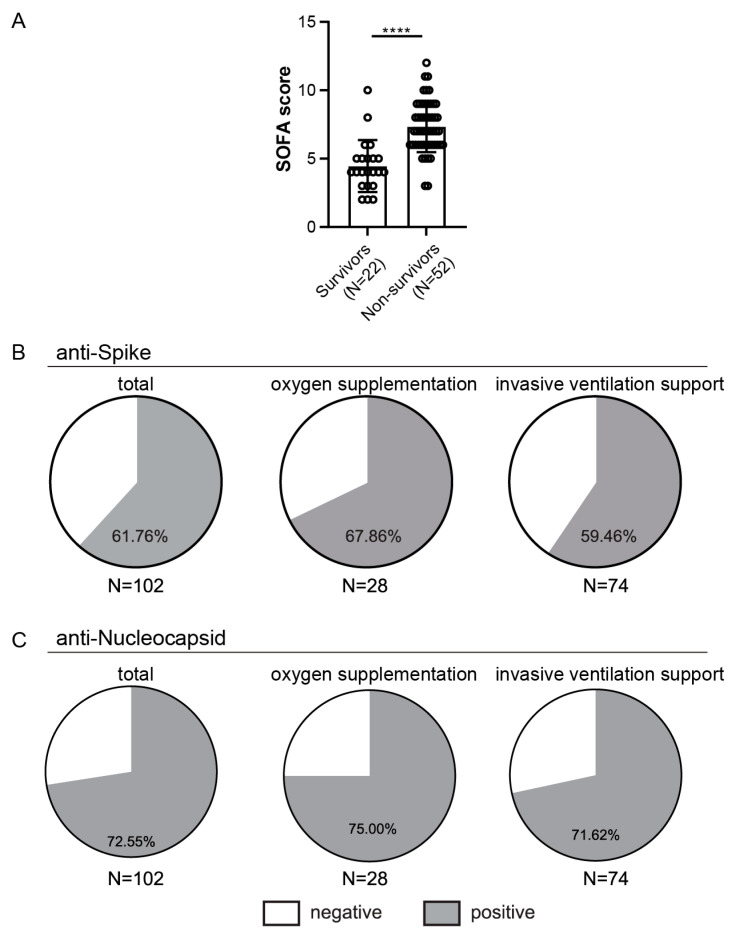
Outcome in relation to SOFA scores and rates of seroconversion in hospitalized patients. (**A**) Comparison of the SOFA scores between survivors and non-survivors from the invasive ventilation support (IVS) cohort. (**B**) Presence of anti-Spike or (**C**) anti-Nucleocapsid antibodies above cut-off value, **** *p* < 0.0001.

**Figure 6 viruses-14-01966-f006:**
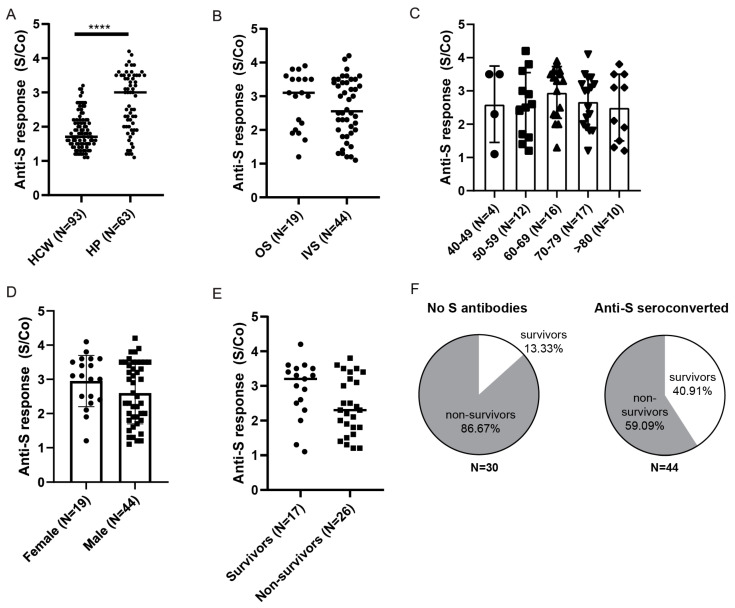
Anti-Spike antibody response in HPs. (**A**) Comparison of anti-Spike antibody levels between in hospitalized patients (HP) and healthcare workers (HCW) identified prior to vaccination. (**B**) Hospitalized seroconverted patients were stratified based on the severity of the disease: oxygen supplementation (OS) vs. invasive ventilation support (IVS), (**C**) age, (**D**) gender, or (**E**) mortality outcome for non-vaccinated patients on invasive ventilation support. (**F**) Frequencies of survivors and non-survivors in relation to anti-Spike seroconversion in IVS cohort. S/Co, Signal to cut-off ratio. Statistical tests: unpaired *t*-test (**A**), *p* value **** *p* < 0.0001.

**Figure 7 viruses-14-01966-f007:**
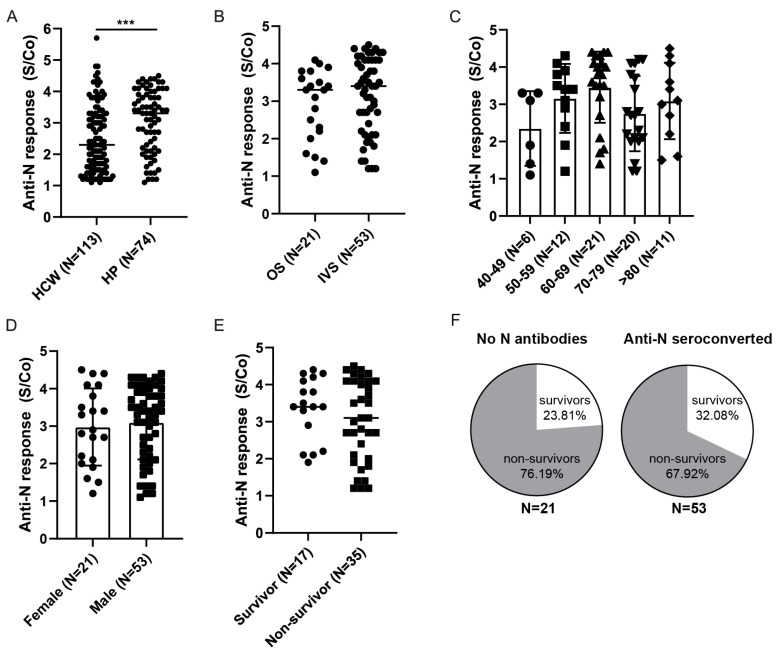
Anti-Nucleocapsid antibody response in HPs. (**A**) Comparison of anti-Nucleocapsid antibody levels between hospitalized patients (HP) and healthcare workers (HCWs) identified prior to vaccination. (**B**) Hospitalized patients were stratified based on the severity of the disease: oxygen supplementation (OS) vs. invasive ventilation support (IVS), (**C**) age, (**D**) gender, or (**E**) outcome for patients on invasive ventilation support. (**F**) Frequencies of survivors and non-survivors in relation to anti-Nucleocapsid seroconversion. Statistical tests: unpaired *t*-test (**A**), *p* value *** *p* < 0.0005. S/Co, Signal to cut-off ratio.

**Table 1 viruses-14-01966-t001:** Demographic characteristics and vaccination status of the studied HCW cohort.

Time Point	Prior Vaccination	+3 Weeks (after 1st Dose)	+6 Months (after 2 Doses)	+14–16 Months
Total number of participants	937	651 (601 *)	380 (291 *)	20 (19 *)
Self-reported COVID-19	104	nd	53	7
Booster dose	/	/	/	14
Gender	77.2% F 22.8% M	75.3% F 24.7% M	78.4% F 21.6% M	70.0% F 30.0% M
Age (mean ± SD)	46 ± 11	45 ± 11	46 ± 11	49 ± 13

* number of patients from the initial cohort (prior vaccination); nd—not determined.

**Table 2 viruses-14-01966-t002:** Demographics of hospitalized patients.

Patient Characteristics			Oxygen Supplementation	Invasive Ventilation Support
		*n* = 102	*n* = 28	*n* = 74
Median age		68 (40–89)	71	67
Gender				
	Female	29 (28%)	8 (28%)	21 (28%)
	Male	73 (72%)	20 (72%)	53 (72%)

**Table 3 viruses-14-01966-t003:** Outcome and patient characteristics for patients on IVS.

Patients on Invasive Ventilation (IVS)		Survivors	Non-Survivors
	*n* = 74	*n* = 22 (30%)	*n* = 52 (70%)
Vaccinated	2/74		
Average time on IVS (days)	10 (±9)	11 (±14)	10 (±7)
Average ICU stay (days)	13 (±11)	18 (±17)	11 (±7)
SOFA score at admission	6 (±2)	4 (±2)	7 (±2)
Gender			
Female	21 (28%)	6	15
Male	53 (72%)	16	37

## Data Availability

All relevant data are available in the manuscript or within the Appendix A.

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
