# Peer review of "SARS-CoV-2 Spike and Nucleocapsid Antibody Response in Vaccinated Croatian Healthcare Workers and Infected Hospitalized Patients: A Single Center Cohort Study"

_viruses, 2022, doi:10.3390/v14091966_

Round 1
Reviewer 1 Report
In this paper, Brlic et al. show data relevant for two groups: health care workers and hospitalized COVID-19 patients. This work shows the binding of serological samples derived from said groups to different strains of COVID-19. The assay described is elegant and scientifically sound. The sample size is large which gives more confidence in the described results and conclusions. And overall the work is well described and the conclusions are clear. We believe that this paper provides an important observation on the human humoral immune response following exposure to SARS-CoV-2 antigens. Thus, this paper is suitable for publication in the Viruses Journal with only minor modifications.
We have two minor points which we think would provide a complementary view to the paper.
First, we think that in Figure 5., Figure 6. And Figure 7. It would be informative to present the results of SOFA scores at admission, or the maximal SOFA score during the disease, as compared to presenting the final outcome of the disease alone. This may help to derive clinically relevant information.
Secondly, we recommend elaborating in the discussion about similar results that were previously reported in literature and provide more explanations for any conflicting results which arise. The authors do write on that in the discussion but this part should be further elaborated.
Author Response
Please find our responses to both reviewers in the attached PDF file "Kucan Brlic et al response to reviewers".

Reviewer 2 Report
Kučan Brlić et al. measured anti-SARS-CoV-2 Spike (S) and Nucleocapsid (N) antibody response in cohorts of healthcare workers and infected hospitalized patients using their in-house developed ELISA assay for up to 16 months after initial sampling. They reported that healthcare workers had a high level of anti-S antibody response. The response was sustained in vaccinees with breakthrough infections or those who received the third, boost vaccine dose. In contrast, N antibody responses were waning faster during the 6-month observation period. Further, they found that infected hospitalized COVID-19 patients had higher levels of anti-S and anti-N antibodies compared to the vaccinees.
The manuscript is well written and logically structured into paragraphs that allow an understanding of the data. The introduction gives a good overview of relevant papers and a detailed discussion is well developed and connects the presented data to several recently published papers investigating anti-SARS-CoV-2 antibody levels in divergent populations. However, there are a few issues, which addressing would improve the quality of the manuscript.
Major comments
1. The authors used self-developed N- and S-protein ELISA to monitor the antibody responses in healthcare workers for up to 16 months. The authors mentioned that the N-protein response can be used “incidence of infected HCWs prior to vaccination and the kinetics of infection-induced anti-Nucleocapsid antibody response”. However, they have not used those results to stratify the cohort into subgroups when monitoring the S-protein-specific antibodies. I would propose that the results for the N-protein antibody response are used in addition to the results of the self-reported questionnaire to stratify the cohort into 14 subgroups: (i) donors with no evidence of infection at t0; (ii) donors with prior contact with SARS-CoV-2 at t0; (iii) donors from the group (i) without evidence of infection at t1; (iv) donors from a group (i) with evidence of infection at t1; (v) donors from a group (ii) at t1; (vi) donors from (iii) without evidence of SARS-CoV-2 infection at t2; (vii) donors from (iii) with evidence of infection at t2; (viii) donors from (vi) at t2; (ix) donors from a group (v) at t2; (x) donors from a group (vi) without evidence of infection at t3; (xi) donors from a group (vi) with evidence of infection at t3; (xii) donors from (vii) at t3; (xiii) donors from a group (viii) at t3; (xiv) donors from a group (ix) at t3. This would allow better distinction of the effects of vaccination on one hand and the combination of infection and vaccination on the other side. Please note that this would also require presenting an anti-N-protein antibody response in Figure 1.
2. At the moment, the data for serum antibody binding to Omicron BA.1, BA.2, and RBD are presented as a percentage of the antibody binding to the original virus variant for each patient. However, this precludes the correlation of anti-original to the Omicron S-protein response. Would it be possible to show the correlation of anti-original and the anti-Omicron anti-S response from different donors stratified into subgroups at t2 as proposed above?
3. Would it be possible to compare the level of antibodies at t3 between donors who were 3x vaccinated and donors that were 3x vaccinated with a previous history of COVID-19 infection between t2 and t3? This could provide information on how much breakthrough infections contribute to the protection and whether the antibody responses boosted by the infection can be further increased by the third vaccination.
4. It would be crucial to add information at which time-point post symptom onset were the samples collected from hospitalized patients. Additionally, would it be possible to compare the antibody titers from the patients infected at different periods, which would be most likely infected with divergent SARS-CoV-2 variants of concern?
5. Was the survival of the COVID-19 patients that did not develop anti-S or anti-N antibodies worse than the survival of patients in whom antibodies could be detected? Moreover, was there a correlation between N- and S- antibody responses in samples from COVID-19 patients?
Minor comments:
1. Please indicate the mean group value for serum antibody binding to Omicron BA.1, BA.2, and RBD in Figures 3B, 3C, and 3E, respectively.
2. Please include information on the type of COVID-19 vaccine used for vaccination in the material and methods. At the moment it is unclear if all donors received only the BNT162b2 vaccine and whether the same vaccine was used for all three vaccinations.
Author Response

(The authors gave the same response as above.)
